# Comparison of Phytochemical Composition and Untargeted Metabolomic Analysis of an Extract from *Cnidoscolus aconitifolius* (Mill.) I. I. Johnst and *Porophyllum ruderale* (Jacq.) Cass. and Biological Cytotoxic and Antiproliferative Activity In Vitro

**DOI:** 10.3390/plants12101987

**Published:** 2023-05-15

**Authors:** Ángel Félix Vargas-Madriz, Ivan Luzardo-Ocampo, Ulisses Moreno-Celis, Octavio Roldán-Padrón, Jorge Luis Chávez-Servín, Haydé A. Vergara-Castañeda, Mónica Martínez-Pacheco, Carmen Mejía, Teresa García-Gasca, Aarón Kuri-García

**Affiliations:** 1Department of Cell and Molecular Biology, School of Natural Sciences, Universidad Autonoma de Queretaro, Querétaro 76230, Mexico; angel.vargas@uaq.mx (Á.F.V.-M.); ulisses.moreno@uaq.mx (U.M.-C.); octavio.roldan@uaq.mx (O.R.-P.); jorge.chavez@uaq.mx (J.L.C.-S.); monicalmp@iibiomedicas.unam.mx (M.M.-P.); maria.c.mejia@uaq.mx (C.M.);; 2Tecnologico de Monterrey, The Institute for Obesity Research, Ave. Eugenio Garza Sada 2501 Sur, Monterrey 64849, Mexico; ivan.8907@gmail.com; 3Tecnologico de Monterrey, School of Engineering and Science, Campus Guadalajara, Av. General Ramon Corona 2514, Zapopan 45201, Mexico; 4Advanced Biomedical Research Center, School of Medicine, Universidad Autonoma de Queretaro, Querétaro 76010, Mexico; hayde.vergara@uaq.mx; 5Laboratorio de Biomedicina Interdisciplinaria, School of Natural Sciences, Universidad Autonoma de Queretaro, Querétaro 76230, Mexico

**Keywords:** *Cnidoscolus aconitifolius*, antiproliferative effect, colorectal cancer, metabolomic analysis, phenolic compounds, phytochemicals, plant extracts, *Porophyllum ruderale*

## Abstract

*Cnidoscolus aconitifolius* (CA) and *Porophyllum ruderale* (PR) are representative edible plants that are a traditional food source in Mexico. This research aimed to analyze the phytochemical composition and untargeted metabolomics analysis of CA and PR and evaluate their antiproliferative effect in vitro. The phytochemical composition (UPLC-DAD-QToF/MS-ESI) identified up to 38 polyphenols and selected organic acids that were clustered by the untargeted metabolomics in functional activities linked to indolizidines, pyridines, and organic acids. Compared with PR, CA displayed a higher reduction in the metabolic activity of human SW480 colon adenocarcinoma cells (LC_50_: 10.65 mg/mL), and both extracts increased the total apoptotic cells and arrested cell cycle at G0/G1 phase. PR increased mRNA *Apc* gene expression, whereas both extracts reduced mRNA *Kras* expression. Rutin/epigallocatechin gallate displayed the highest affinity to APC and K-RAS proteins in silico. Further research is needed to experiment on other cell lines. Results suggested that CA and PR are polyphenol-rich plant sources exhibiting antiproliferative effects in vitro.

## 1. Introduction

Traditional plants have been widely used in Mexico as an alternative food source providing additional health benefits. Among these plants, “Quelites”, derived from the Nahuatl word “quilitl”, group a wide variety of up to 250 different plants consumed as a food product or either as medicinal plants in Mexico and Central America, and includes shrubs and trees providing leaves, tender stems, and inflorescences [1]. Despite their cultural, nutritional, and medical use, fewer than 30 species have been properly documented for their proximal and nutraceutical composition and their biological properties in vitro and in vivo [2]. Of note, most of their potential biological properties have been linked to their polyphenolic content, as most of these components have been widely associated with many biological benefits [3].

The main “Quelites” consumed in Mexico and Central America are *Cnidoscolus aconitifolius* (Mill.) I. I. Johnst (Euphorbiaceae) (CA) and *Porophyllum ruderale* (Jacq.) Cass. (Asteraceae) (PR) [1,4]. In traditional medicine, CA leaves are used to treat different inflammatory diseases such as cancer, arthritis, diabetes mellitus, and gastrointestinal problems [5]. Moreover, PR is used as an anti-inflammatory treatment of lymph nodes and is valued for its antiparasitic properties [1,6]. Together with their nutritional properties and use in food preparations, the plants have been considered for the treatment of gastric conditions, ulcers, vomiting, hemorrhoids, dysentery, colic, and indigestion [7]. Chronic inflammation has been shown to be a risk factor for developing colorectal cancer (CRC) [8]. However, information regarding its biological properties in disease models is scarce, and there have not been examinations of its anticancer effects. The CA food and medicinal value have been widely explored due to its efficacy in treating several chronic conditions as an antidiabetic, antibacterial, hepatoprotective, anti-inflammatory, and analgesic [4]. For colon adenocarcinoma, particularly, CA root bark extracts have only been tested in vitro in advanced stages (human HT-29 cells). In contrast, leaf extracts have been assayed in colon cancer in vivo [9], displaying inhibition of the cancer cell proliferation of early neoplastic lesions and reduction in colonic inflammation through the involvement of β-catenin. However, additional intermediaries involving *Apc* and *Kras* genes have not yet been explored. In this sense, research considering natural products such as edible plants continues to gain relevance in the search for new food products with additional health benefits. Therefore, this research aimed to elucidate the phytochemical composition and conduct an untargeted metabolomic analysis of edible “Quelites” such as *Cnidoscolus aconitifolius* (Mill.) I. I. Johnst and *Porophyllum ruderale* (Jacq.) Cass. and evaluate their biological antiproliferative activity in vitro.

## 2. Results

### 2.1. Polyphenol Profile and Untargeted Metabolomics from CA and PR Samples

Table 1 shows the total amount of free total phenolic compounds (TPC), total flavonoid content (TFC), and condensed tannins (CT) of CA and PR samples. The CA samples were significantly higher (*p* < 0.05) for TPC and TFC compared with PR, but PR displayed a higher amount (+25.03%, *p* < 0.05) of CT in comparison to CA.

Appendix A shows the polyphenolic, selected acids, and other compounds profile of both CA and PR extracts by UPLC-DAD-QToF/MS-ESI. Identified compounds were classified into 7 classes such as hydroxycinnamic acids and derivatives (8 compounds), benzoic acid derivatives and benzaldehydes (6 compounds), flavonols (7 compounds), flavones (9 compounds), flavanones (2 compounds), phenylpropanoids (1 compound), and other compounds (4 compounds). Out of 38 identified compounds, CA contained 24 compounds and PR 30 compounds, most of them hydroxycinnamic acids (CA: 5 and PR: 6 compounds), flavones (CA: 5 and PR: 7 compounds), and flavonols (CA: 5 and PR: 7 compounds).

The quantification of selected phenolic compounds is presented in Table 2, where CA displayed the highest abundance of compounds, particularly hydroxycinnamic acids and caffeic acid being remarkably higher (*p* < 0.05) (10.02-fold) than PR, but PR showed a higher number of flavones, particularly (+)-catechin, epicatechin, and epigallocatechin gallate, compared with CA (*p* < 0.05). Despite an overall lower number of compounds, certain compounds were only detected and quantified in PR, such as resveratrol, quercetin, hydroxyphenylacetic, and hydroxybenzoic acids, some of them confirmed in Appendix A. Representative chromatograms of some of the identified compounds in the positive and negative electrospray ionization (ESI) are presented in Appendix A and Appendix A, respectively.

An untargeted metabolomic profile was carried out for the identified phenolics by UPLC-DAD-QToF/MS-ESI (Figure 1), where the highest relative abundances were shown for CA extracts, except for sinapic acid, quercetin, and rutin, which presented a lowest relative abundance (Figure 1A). Despite selected differences for each compound, all CA and PR samples were clustered within the same groups, indicating a compositional homogeneity. Further analysis of the variable in importance (VIP) scores (Figure 1B) suggested that only Kaem-3 (kaempferol-3-*O-*rutinoside or nicotiflorin) showed VIP ≥ 2, whereas all the other compounds could not reach this significant boundary. The confirmation of the abundance clustering of CA and PR samples is shown in the principal components analysis (PCA) of both extracts (Figure 1C), where a complete separation between both groups is shown, and the variance is explained with the exception of two components (both reaching 99.8% of the total difference between samples). For the Mummichog pathway and network analysis (Figure 1D), the prediction of functional activity targeted up to 31 compounds’ classifications related to the chemical profile of CA and PR metabolites. Overall, CA samples displayed a major richness of compounds, where indolizidines, quinoxalines, benzopyrans, and organic dicarboxylic acids presented the highest prevalence. Regarding PR, indolizidines, dialkyl ethers, amino acids and peptides, pyridines, organic sulfuric acids, and organic thiophosphoric acids were the most prevalent classes.

### 2.2. CA and PR Extracts Decreased SW480 Metabolic Activity in a Dose–Response Manner

The results for the metabolic activity behavior at several CA and PR concentrations are presented in Figure 2, where a dose–response trend was shown for both CA (Figure 2A) and PR (Figure 2C) extracts, although a higher PR concentration was needed to achieve similar results than CA. Quantification of the half-lethal concentration (LC_50_) showed that CA extracts are more potent than PR (190.5-fold) (Figure 2B,D).

### 2.3. CA and PR Extracts Induced SW480 Apoptosis and Arrested Cell Cycle at G0/G1 Stage

Examination of the SW480 cell death and lactate dehydrogenase release after treatments with the LC_50_ doses of CA and PR extracts are shown in Figure 3. Representative pictures plotting cell viability vs. Annexin V screening for the untreated cells, CA, and PR (Figure 3A) and their quantification (Figure 3B) indicated that despite the high percentage of cells that remain alive after the treatments (34.85 and 44.62% for CA and PR, respectively), both treatments are effective in inducing apoptosis, CA-treated cells were mainly at a late apoptotic process (30.5%), whereas PR-induced cells exhibited mostly an early apoptotic stage (35.9%). For the lactate dehydrogenase release (Figure 3C), only PR extracts showed a level similar to the untreated control (*p* > 0.05).

Cell cycle evaluations were conducted to deepen the understanding of the apoptotic process. Representative histograms of the cell cycle distribution based on the DNA content index and the number of counts (Figure 3D) and the quantification of total cells for each cell cycle step (Figure 3E) revealed that most cells are in the G0/G1 stage; no differences were presented at the S stage between all treatments, and significantly less (*p* < 0.05) CA and PR-challenged cells were in the G2/M stage, compared with the untreated SW480 cells.

### 2.4. Impact of CA and PR Extracts on Apc and Kras Gene Expression

The relative mRNA gene expression of the *Apc* and *Kras* genes is presented in Figure 4. The total quantification of the mRNA *Apc* expression (Figure 4A) showed a higher PR ability to increase *Apc* expression compared with untreated or CA-treated cells (*p* < 0.05) (5.94- and 0.52-fold, respectively, for control and CA-challenged cells). There were no differences (*p* > 0.05) in the gene expression after LC_50_-CA or LC_50_-PR treatments (relative expression: ~0.05–0.06) (Figure 4B), although both were lower than the untreated control. Examination of potential in silico binding interactions between CA and PR’s representative phenolics against the codified protein for *Apc* showed that among all the examined interactions (Appendix A), rutin showed the best energy binding affinity for both extracts (ΔG: −8.50 and −8.70 kcal/mol for APC and K-RAS protein, respectively) (Figure 4C,D). Most interactions were dominated by conventional hydrogen bonds, carbon–hydrogen bonds, and pi alkyl. Regarding the amino acids involved in the interactions with the targeted phenolics (Appendix A), rutin interacted with eight amino acids. Additional energy binding affinities are shown in Appendix A and Appendix A. Compared with the obtained results for CA, a major diversity of interactions was found for PR, particularly for additional pi-cation and pi-anion, and pi-donor hydrogen bonds.

### 2.5. Principal Components Analysis (PCA) for the Evaluated Responses in SW480 Cells

Taken together with all results, Figure 5 shows the PCA analysis of the evaluated responses in SW480 cells after CA and PR treatments. Up to three components explained >80% total variance (Figure 5A). Considering the participation of each variable in the PCA analysis (Appendix A), the separation of CA and PR samples in Figure 5B is mainly explained by differences between their polyphenolic content, their impact on the cell viability, and the number of cells in the G0//G1 stage. Moreover, it could be observed that the total polyphenolic content impact early apoptosis, whereas epigallocatechin gallate and hydroxybenzoic acids are positively related to *Apc* gene expression, resveratrol, and vanillin contents with live cells; and condensed tannins and caffeic acids with the induction of early apoptosis.

## 3. Discussion

Cancer is a multifactorial heterogeneous metabolic pathology, defined as an irreversible alteration in cell homeostasis [10]. Colorectal cancer (CRC) is currently the third leading cause of cancer death, and global mortality cases are estimated to increase by 60% by the year 2030 [11]. Although several treatments have been proposed for CRC, most of these procedures involving pharmacological methods display several side effects that could be reduced with the involvement of natural products aiming to either reduce the effect of these treatments or diminish the risk of having CRC [12]. Genetic and epigenetic alterations are involved in CRC as well as mutations that inactivate the function of the *Apc* gene and the *Ras* oncogene, mainly in KRAS. The latter oncogene contains membrane-associated-GTPases that are involved in cell survival, proliferation, and differentiation. In CRC, mutations in the *Kras* oncogene and those of the *Apc* gene lead to greater cell proliferation, the MAPK signaling caused by mutated KRAS presents ERK hyperphosphorylation, which activates different effector mechanisms such as the G1/S phase transition, and the inhibition of apoptosis. The defective APC, coupled with the mutated KRAS, inhibits the activity of GSK3β (glycogen synthase kinase) and amplifies the activity of β-catenin [13].

Among the natural treatments that could be used as a chemopreventive approach in CRC treatment, natural plants have emerged as a hot topic due to the revalorization of their phytochemical composition, exhibiting various biochemical mechanisms inhibiting CRC progression [14]. “Quelites” groups a wide variety of plants, used either as a food complement or medicinal plants, that also contribute to providing food security since most of these plants are traditionally cultivated in home gardens or “milpas”, which also offer a high capacity of biodiversity conservation [15]. Overall, “Quelites” have been reported to be a proper source of macromolecules such as protein, amino acids, minerals (Ca, Mg, and Zn), vitamins (C and E), and fiber, but also bioactive components like phenolic acids (particularly caffeic and ferulic acids), flavonoids (quercetin, kaempferol, and spinacetin), carotenoids, and fatty acids (α-linoleic acid) [16,17]. For *C. aconitifolius* particularly, its fiber (1.9 g/100 g fresh weight), P (39 mg/100 g), K (217.2 mg/100 g), Fe (11.40 mg/100 g), and ascorbic acid (164.7 mg/100 g) contents are higher than spinach, which is a commonly used plant to compare CA culinary uses and nutritional properties [18]. The results obtained in this research from TPC and TFC are similar to previously published reports for similarly prepared CA extracts (TPC: 5250 mg GAE/100 g and TFC: 416 mg CE/100 g) [9] but are higher than the values reported by Babalola and Alabi [19] (TPC: 2550 mg GAE/100 g TFC: 183.33 mg/100 g dry weight) and (TFC: 33.29 mg CE/100 g fresh weight) [20]. Regarding PR, TPC contents reported in this research are higher than the values indicated by [3] (TPC: 316–391 mg GAE/100 g), although the authors prepared a 2:1 aqueous extract without stirring, which could influence the yield of released components.

The identified phenolics in CA are similar to those previously reported [21,22], but the values for quercetin are lower than those informed (16.9 µg equivalents/g sample) [21,22]. Quercetin, kaempferol, rutin, protocatechuic acid, and glycosilated flavonols are some of the most reported phenolic compounds in ethanolic and methanolic extracts of CA [23]. The stated phytochemical composition from this research is higher than values found for wild and cultivated *P. ruderale* (75% v/v ethanol extraction of 2 g leaves, followed by 2 h agitation) [3], but the authors also reported chlorogenic, caffeic, and *p-*coumaric acids; rutin and quercetin, and also indicated contents of gallic acid, myricetin, luteolin, kaempferol, and apigenin that were not found in this research. Similarly, an extensive review of the phytochemical composition of several native plants from Brazil indicated that, among *P. ruderale* main phenolics, gallic and quinic acids, quercetin, and rutin could be found [24]. However, quinic acid and myricetin were identified in PR extracts after UPLC-DAD-QToF/MS-ESI analysis. The identification of compounds by liquid chromatography coupled with QTof mass spectrometry in a 40% ethanol solution reported by Athayde et al. [20] indicated the presence of several compounds also found in this research, such as quinic acid, chlorogenic acid, quercetin (*p-*coumaroyl), rutin, and chlorogenic and caffeic acids derivatives. It has been concluded that together with the representative polyphenolic classes indicated in this research, a class of nitrogenated compounds (e.g., C_58_H_64_N_3_O_10_, C_58_H_62_N_3_O_10_) of PR are some of the major contributors to its phytochemical variability, as these compounds have been depicted as specific to PR and other local *Asteraceae* plant species [25]. The polyphenolic composition of CA and PR explained by the Mummichog pathways after the untargeted metabolomic analysis, primarily depicted indolizines, quinoxalines, organic disulfuric acids, and dicarboxylic acids as the main involved pathways, showing the plasticity of the phenylpropanoids pathways to provide the richness of CA and PR phytochemicals [26]. Unfortunately, a metabolomic biosynthetic approach for these plants considering agronomic features has yet to be reported, and more research is needed to elucidate their phytochemical composition for future biotechnological applications.

Due to the richness of biologically active compounds, the potential of “Quelites” to reduce the risk of chronic non-communicable diseases has been explored, including cancer. Despite several claims of the ability of *C. aconitifolius* to be used in cancer treatment [27], only a few reports have currently demonstrated this potential through in vitro and in vivo mechanisms. To our knowledge, there are no reports of the in vitro activity of CA extracts on colon cancer. However, our research group previously reviewed the impact of *C. aconitifolius* consumption in an azoxymethane-induced early colorectal cancer in vivo model [9]. In this research, colon cancer was induced using azoxymethane, a common carcinogen, in male Sprague–Dawley rats, and CA was administered to the rats as an aqueous extract (10 g of the leaves boiled in 1 L water for 5 min). Despite no differences shown for the body weight evolution of the animals between the tested groups, a significant reduction (−29.5 to −64.6%) in colonic aberrant crypt foci in a chronic and sub-chronic administration of the leaves (16 and 32 weeks, respectively) was found, suggesting the effect of *C. aconitifolius* bioactive components in reducing the colonic histological alteration. No individual compounds from CA were evaluated in this model, but the CA phytochemical composition, predominantly governed by *p-*coumaric, rosmarinic, and chlorogenic acids, was linked to the chemopreventive effects in reducing the immunohistochemical expression of proteins in the colonic tissue linked to CRC progression and development such as β-catenin, proliferating cell nuclear antigen (PCNA), caspase-3, cyclo-oxygenase-2 (COX-2), and the nuclear factor kappa B (NF-κB). Yet, no differences were found between the CA and AOM + CA treatments, suggesting that additional mechanisms could be explored to explain the chemoprotective effects of CA. In another study, for both breast (MCF-7) and lung (NCI-H460) cancer cell lines, methanolic CA extracts (1–250 µg/mL) significantly reduced cell growth up to 77.68% in the lung cancer cells, displaying LC_50_ > 250 µg/mL, which is lower than the values reported in this research (4470 µg/mL), although biological features of the tested cell lines could explain these differences [28]. In this research, PR extracts showed higher (*p* < 0.05) LC_50_ values than CA. According to the criteria of the National Cancer Institute and the Geran protocol, extracts with IC_50_ ≤ 20 µg/mL are highly cytotoxic, IC_50_ between 21 and 200 µg/mL are moderately cytotoxic, IC_50_ between 201 and 500 µg/mL are weakly cytotoxic, and IC50 > 501 µg/mL are not cytotoxic [29]. Based on the described criteria, PR is classified as non-cytotoxic, and CA is slightly cytotoxic. The results of different studies using extracts of medicinal plants in different cell lines have shown IC_50_ values lower than those reported in the present study [30]. However, this may be due to different factors, such as the nature of the bioactive compounds of the plant that may be varied due to the environmental and growing conditions of the location where they are found, as well as the sensitivity of the different cell lines to the extracted bioactive compounds. [31]. Furthermore, the choice of solvent to use for the extraction of bioactive compounds can influence the cytotoxic effect of the cell line; commonly used solvents are methanol, ethanol, hexane, and mixtures of these [30]. For example, the study carried out by Godínez–Santillán et al. [32] used methanol and ethanol in proportions of 80:20 and 50:50 v/v, obtaining more phenolic compounds than the present study. On the other hand, in traditional medicine, water is used to carry out the extractions. However, the drawback is that this solvent primarily extracts polar compounds, and it has also been observed that the aqueous extract of a single plant commonly presents poor cytotoxicity. However, aqueous extraction with different plants has been observed to have stronger cytotoxic effects [30]. In this way, it can be suggested that the concentrations used in the cell lines to observe a cytotoxic effect can vary depending on the concentration of the bioactive compounds extracted from the plants [31].

In this research, PR extracts showed higher (*p* < 0.05) LC_50_ values, but even less information on the anti-cancer potential of *P. ruderale* is found in the literature. The administration of *P. ruderale* hydroalcoholic solution intraperitoneally (100 and 200 mg/kg body weight) for 10 days in B16F10 murine melanoma cells treated C57BL/6 mice showed an impact in the animals’ weight (PR: 200 mg/kg), but no differences were found in the liver weight. Compared to a control (doxorubicin), PR groups displayed reduced liver protein deposition, preservation of the liver’s architecture, and reduced metastatic process, confirming the chemoprotective effects of PR extracts [33].

The ability of PR and CR extracts to induce apoptosis, induce LDH release, and arrest the cell cycle was confirmed in this research as the LC_50_ concentrations of the extracts significantly increased the number of cells in total apoptosis, but PR treatment was more effective. In contrast, CA was more cytotoxic, inducing higher levels of LDH release. Despite no available reports for CA and PR inducing LDH release and apoptosis, medicinal leaves have displayed the reduction in LDH release, an indicator of cell membrane damage [34], in the same tested cell line of this research. For instance, Shang et al. [35] indicated anti-proliferative properties of methanolic extracts *Luffa echinata* Roxb. (*Cucurbitaceae*) fruits on SW480 cells (70.2 µg/mL) after a 24 h exposure and moderate LDH releases (up to ~3–4% of total LDH), which are much lower than those found in this research. Unfortunately, the authors did not link the LDH release with observed pro-apoptotic or cytotoxic outcomes but based on the chemical composition of the tested leaves, most of these results could be linked to glycosylated terpenes and triterpenes. Digested extracts of *Moringa oleifera,* a well-known plant with food and medicinal properties, displayed pro-apoptotic and necrotic effects on human HT-29 colorectal cancer cells [36]. The extracts showed a high abundance of antioxidant polyphenols such as gallic, chlorogenic, caffeic, ellagic, *p-*coumaric acids, and flavonoids like rutin, (+)-catechin, morin, quercetin, and kaempferol, that might be exerting pro-apoptotic and cytotoxic effects in the colon cancer cells. It is worth noting that apoptosis is a complex process involving the activation of several proteins and biochemical mechanisms that cannot be wholly elucidated through the cell cycle and apoptosis measurement through colorimetric or fluorometric methods. However, the provided quantifications from this research open the possibility of further exploring these mechanisms through additional proteomics and gene expression analysis.

The normal cell cycle develops according to the G1-S-G2-M phases, and in this cellular mechanism, the synthesis and distribution of nucleic acids, chromosomes and proteins are verified. At an abnormal point, the cell cycle proceeds disorderly, causing alterations in cell proliferation and differentiation [37]. Cell cycle arrest is probably one of the most successful strategies in CRC treatment, as cyclin-dependent kinase mutations highly influence CRC development and progression. Treatments showed that most cells were in a G0/G1 stage, followed by S, and G2/M, indicating an arrest preventing SW480 cells from entering the S phase, which is a clear indication of phytochemicals inducing DNA damage in the cells [38]. Before the G0/G1 arrest, rapid induction of cyclin-dependent kinase inhibitors occurs, leading to phosphorylation of pocket protein family members, inhibiting cell cycle progression [39]. Phenolics from medicinal plant extracts such as *p-*coumaric, protocatechuic, and caffeic acids, some of which were reported for CA and PR, are apoptotic inductors elevating caspase-3 levels, reducing membrane potential, and inducing morphological changes in breast, prostate, lung, cervical, fibrosarcoma, and colorectal cancer cells [40,41]. Since PR induced a major accumulation of cells in the sub-G0/G1 phase, it can be suggested that PR is a more potent apoptosis inductor in SW480 cells than CA [42].

Colon cancer is a multistage disease characterized by successive changes in various genes, including *Apc* and *Kras*, leading to changes in CRC signaling pathways such as the APC Wnt/β-catenin and K-RAS [43]. As *Apc and Kras* are frequently mutated in CRC, targeting these genes could provide insights into the effects of bioactive compounds in CRC treatment [44]. Exposure to PR extract increased the *Apc* gene, while the opposite was presented in CA treatment. Since APC is involved with β-catenin transport out of the nucleus to allow its phosphorylation and degradation, the pro-apoptotic PR effects in SW480 cells might be related to other mechanisms that should be further explored. In the present study, a significant decrease in the relative expression of the *Kras* mRNA oncogene was observed by both extracts, and the in silico analysis highlighted rutin as one of the phenolics with major interaction with the protein codified for this gene. Rutin has been proposed as a cell signaling modulator in MAPK signaling, reducing the expression of the death receptors-4 and -5 (DR4/DR5), Akt, ERK, and NF-κB in vitro and in vivo [45].

The overall integration between the polyphenolic composition and the biological properties of the extracts in the PCA analysis suggested that compounds such as epigallocatechin-gallate are involved in *Apc* modulation, which might be inferred due to the high binding affinity between epigallocatechin gallate and APC protein, whereas pro-apoptotic effects could be associated to caffeic acids, condensed tannins, total phenolic compounds, and gallic acids, which are known anti-inflammatory compounds and reactive oxygen species inductors in cancer cells [46]. The individual evaluation of these components directly extracted from CA and PR might explain the contributions of phytochemicals to CRC development in vitro and in vivo.

## 4. Materials and Methods

### 4.1. Plant Materials

Fresh leaves of *Cnidoscolus aconitifolius* (Mill.) I. I. Johnst. (Euphorbiaceae) (CA) (World Checklist of Selected Plant Families, WCSP record: 44157) and *Porophyllum ruderale* (Jacq.) Cass. (PR) [The International Compositae Alliance (TICA) checklist record: D78AD427-6E4D-4C8C-8C6A-BF2EDC6EA06C] were obtained from traditional markers of the city of Queretaro (Mexico). Plants records were acquired from the Plant List database (http://www.theplantlist.org, accessed on 2 July 2022). The leaves were identified by a specialist from the Jerzy Rzedowski Herbarium of the Universidad Autonoma de Queretaro and sampled (voucher specimens QMEX8210 and QMEX157).

Once identified and washed with distilled water and disinfected (sodium hypochlorite, 200 ppm), the leaves were dried in a forced hot air oven (Shel Lab FX 1375, Sheldon Manufacturing, Cornelius, OR, USA) at 40 °C for 72 h until a constant weight was obtained. The dried leaves were ground in an electric mill (Thomas Model 4 Wiley Mill, Thomas Scientific, New Jersey, USA) and screened through a 0.5 mm particle-size sieve. The ground powder was collected in an airtight bag and stored at −80 °C for further processing.

### 4.2. Extracts Preparation, Identification, and Quantification of Phenolic Compounds

#### 4.2.1. Extracts Preparation

Powdered CA and PR samples were mixed with distilled water in a 1:10 solid:liquid ratio and stirred (100 rpm, 22 ± 1 °C) for 16 h, protected from light. Solutions were then filtered with a Whatman paper (0.22 μm) and freeze-dried for 72 h (FreeZone 6 Liter Benchtop Freeze Dry System, Labconco, Kansas City, MO, USA). The resulting extracts (lyophilized extract or LE) were stored in tubes protected from light at −80 °C for further analysis.

#### 4.2.2. Quantification of Total Phenolic Compounds, Total Flavonoids, and Condensed Tannins

The total phenolic compounds (TPC) were determined following the Folin–Ciocalteu method [47], and the results were expressed in milligrams equivalents of gallic acid per gram of freeze-dried extract (mg GAE/g LE). The total flavonoid content (TFC) was quantified as indicated by Zhishen et al. [48], and results were expressed in mg (+)-catechin/100 g LE. Condensed tannins (CT) were determined using the vanillin method [49], and the values were also presented as mg (+)-catechin equivalents/100 g LE (mg CE/100 g LE).

#### 4.2.3. Identification of Phenolic Compounds by UPLC-DAD-QTof/MS-ESI

The free phenolic compounds from the prepared extracts were analyzed in an Ultra-Performance Liquid Chromatograph (UPLC) coupled to a diode array detector (DAD) and a quadrupole time-of-flight (QToF) mass spectrometer (MS) with an electrospray ionization (ESI) interphase (Vion IMS, Waters Co., Milford, MA, USA) as reported in Sánchez-Recillas et al. [50] with slight modifications. Briefly, after filtration of the samples (0.22 µm, Acrodisc^®^ Syringe Filters, Agilent Technologies, Palo Alto, CA, USA), injection into an Ethylene Bridge Hybrid (BEH) Acquity C18 column was conducted (2.1 × 100 mm, 1.7 µm granule size) and thermostatically stabilized at 35 ± 0.6 °C. The separation was carried out using water acidified with 0.1% v/v formic acid (A) and acidified acetonitrile with 0.1% v/v formic acid (B), and gradient conditions as follows: 5% B at 0 min, 5% B at 2 min, 95% B at 22 min, 95% B at 25 min, 5% B a 27 min, and 5% B at 30 min. The samples (10 °C) were separated at a flow rate of 0.4 mL/min using an injection volume of 2 µL. An absorbance scanning (210–660 nm) with selected channels (214, 280, 320, and 360 nm; 520 nm for the positive scan mode) was set. Ionization was conducted in positive and negative scan modes. The MS analysis method consisted of the following parameters: low collision energy: 6 eV, 15–45 eV high collision energy, mass range: 50–1800 m/z; capillary voltage: 2 and 3.5 kV for the negative and positive scans, respectively; source temperature: 120 °C, desolvation temperature: 450 °C. Argon was used as collision and desolvation gas at 50 L/h and 800 L/h, respectively, using 40 V as cone voltage. Leucine-enkephalin solution (50 pg/mL) was used to lock mass correction at 10 mL/min. The exact mass of the pseudo molecular ions (mass error < 10 ppm), isotope distribution, and fragmentation patterns were used to identify each compound. Data were analyzed with the UNIFI 1.9 SR 4 software (Waters Co., USA) using at least 5 ppm for targeting compounds. Fragments were identified using the reported fragmentation patterns from PubChem, FooDB v. 1.0, HMDB v. 5.0, and MassBank of North America (MoNA).

#### 4.2.4. Identification and Quantification of Free Phenolic Compounds by HPLC-DAD

The free phenolic compounds were identified and quantified following the procedure of Figueroa et al. [51] with slight modifications. A high-performance liquid chromatography (HPLC) system (Agilent 1100, Agilent Technologies, Palo Alto, CA, USA) coupled to a diode-array-detector (DAD) was used with a Zorbax Eclipse XDB-C18 column (4.6 × 250 mm, 5 μm granule size; Agilent Technologies). The column was thermostatically adjusted (35 ± 0.6 °C), and a flow rate of 1 mL/min was used. Two mobile phases were used: solvent A (acidified water with formic acid, 0.1%) and solvent B (100% acetonitrile). The gradient was set as indicated: 95% A 0 min; 50% A (25 min), 0% A (33 min), and 95% A (36 min). The detection was conducted at 280 nm for phenolic acids and 320 nm for flavonoids with an injection volume of 20 μL. To quantify individual phenolic compounds, standard curves (0–1 mg/mL) of caffeic, chlorogenic, *p-*coumaric, ferulic, 4-hydroxybenzoic, 4-hydroxyphenyl acetic, gallic, and sinapic acids; (+)-catechin, epicatechin, epigallocatechin gallate, quercetin, resveratrol, and rutin. The reference phenolic compounds were HPLC-grade reagents ≥95%, solid (Sigma-Aldrich) and were confirmed based on comparing the corresponding standards’ peak spectrum and retention times. The quantification of the phenolic compounds was carried out using the area of the peaks of the samples concerning the standard curves of the phenolic compounds. The content of phenolic compounds was expressed in mg/mL DW (dry weight), and the analysis was carried out in triplicate.

### 4.3. Untargeted Metabolomic Analysis

A global and untargeted metabolomic analysis was conducted for the identified compounds [52] using the MetaboAnalyst v. 5.0 software [53]. Heatmaps depicting the relative abundance of each compound in the samples were generated by normalizing components by sum and Ward clustering. A Partial Least Squares Analysis (PLSA) using the provided algorithm was used to rank each compound according to the number of components and variables for each model. Rankings were classified using the Variable in Importance (VIP) scores, where a VIP > 2 was considered significant. Principal component analyses (PCA) for the identified compounds were also conducted. An Enrichment Analysis was carried out using the Mummichog Algorithm based on the changes across multiple compounds involved pathways to the target chemical signatures of each sample.

### 4.4. Cell Culture Assays

Human SW480 [SW-480] (ATCC CCL-228) colon cancer cells were obtained from the American Type Culture Collection (ATCC^®^). Cells were seeded in 60 mm diameter dishes in Dulbecco’s Modified Eagle’s Medium (DMEM) (GIBCO, New York, NY, USA), supplemented with 10% fetal bovine serum (FBS) (Biowest, Lakewood Ranch, FL, USA) and 10% Antibiotic-Antimycotic solution (Sigma-Aldrich, St. Louis, MO, USA) at 37 °C under a 5% CO_2_-saturated water atmosphere, with medium changes every two days until 80% confluence was reached.

#### 4.4.1. Determination of Cell’s Metabolic Activity by 3-(4,5-dimethylthiazol-2-yl)-2,5-diphenyltetrazolium Bromide (MTT) Assay

The cells (1.5 × 10^4^ cells/well; 100 μL) were cultured in 96-well plates for 24 h. Then, cells were exposed to serial concentrations of the aqueous extracts (CA: 0.5, 1.0, 2.5, 5.0, 7.5, and 10 mg/mL; PR: 0.1, 0.5, 1, 5, 10, and 50 mg/mL), dissolved in DMEM supplemented with 0.5% bovine seric albumin (BSA), for 24 h (37 °C). After incubation, the medium was replaced with DMEM supplemented with 3-(4,5-dimethylthiazol-2-yl)-2,5-diphenyltetrazolium bromide (MTT) (0.5 mg/mL, 100 μL each well, Sigma-Aldrich) for 4 h (37 °C). The media was then replaced with dimethyl sulfoxide (DMSO), and the absorbance was read at 562 nm. The half-lethal concentration (LC_50_) was calculated using a dose–response equation provided by GraphPad Prism v. 8.0 software. Untreated cells were used as a negative control. Triton X-100 (Sigma-Aldrich) was used as a positive control (1 µL of the pure reagent was dissolved in 100 µL for each well).

#### 4.4.2. Determination of Lactate Dehydrogenase (LDH) Release

The cells (1.5 × 10^4^ cells/well; 100 μL) were cultured as described above. After 24 h, cells were treated with LC_50_ concentrations of CA and PR, while untreated cells (DMEM + 0.5% BSA) were used as a negative control. Then, the LDH cytotoxicity Assay kit (K311–400, Biovision, Milpitas, CA, USA) was used to assess the impact of the treatments on the LDH release following the manufacturer’s instructions. Samples were read at 492 nm, and results were expressed as a cytotoxicity percentage using the equation:% cytotoxicity = [(Sample − Negative control)/positive control] × 100 − Negative Control. Triton X-100 (Sigma-Aldrich) was used as a positive control (1 µL of the pure reagent was dissolved in 100 µL for each well).

#### 4.4.3. Apoptosis Assessment by Flow Cytometry

The Muse^®^ Annexin V and Death Cell Assay Kit (MCH 100105, Millipore, Darmstadt, Germany) was used to assess the impact of CA and PR treatments on the induction of apoptosis in the cells. Briefly, the cells were cultured under proper conditions as described above and after reaching confluence, were treated either with CA- or PR-LC_50_ concentrations. After harvesting by trypsinization and concentration by centrifugation (6000× *g*, 5 min), cells were washed with 1 mM phosphate-buffered saline solution (PBS), adjusted to 1 × 10^6^ cells/mL, and determination was conducted in a Guava^®^ Muse^®^ Cell Analyzer (Luminex, Austin, TX, USA). Results were shown in the percentage of live, early apoptotic, late apoptotic, or total apoptotic cells (%). Untreated cells (DMEM + 0.5% BSA) were used as a negative control.

#### 4.4.4. Cell Cycle Analysis by Flow Cytometry

The cells (3 × 10^5^ cells/60 mm-dish) were treated with CA- or PR-LC_50_ concentration for 12 h, and were then collected by trypsinization and centrifugation (6000× *g*, 5 min), washed with 1 mM PBS + EDTA and fixed in 70% v/v ethanol for 4 h at −20 °C, following the manufacturer’s conditions (MCH1006, Muse^®^ Cell Cycle Assay Kit, Millipore, Darmstadt, Germany) in a Muse^®^ Cell Analyzer (Luminex, Austin, TX, USA). Untreated cells (DMEM + 0.5% BSA) were used as a negative control.

#### 4.4.5. Evaluation of *Apc* and *Kras* Gene Expression by qPCR

Cells were cultured and treated as indicated above. The extraction and purification of mRNA were carried out by adding 400 μL of Trizol (Invitrogen™, Carlsbad, CA, USA. Research, Cat. No. R2052, Irvine, CA, USA) to each dish. The samples were resuspended in nuclease-free water, and total RNA was quantified. Purity was determined using a NanoDrop™ 2000/2000c spectrophotometer (Thermo Scientific, Waltham, MA, USA). Each 2 μg of mRNA was converted for the cDNA synthesis following the kit supplier’s instructions (Maxima H Minus First Strand cDNA Synthesis Cat. No. K1652, ThermoScientific, Waltham, MA, USA) for a reaction volume of 40 μL. The programmed cycle was conducted at 25 °C for 5 min, 65 °C for 30 min, and 85 °C for 15 min. Primers were designed by selecting two gene targets (*Apc* and *Kras*) related to colorectal carcinogenesis and apoptosis pathways (Appendix A). The genetic sequences were analyzed by the UCSC Genome Browser at the University of California, and primers were designed using the Primer3 page, selecting a TM of 60 ± 2 °C (for 20 ± 2 bp), product size of 100–250 bp, and GC ≥ 50%. The sequences of the target genes (Adenomatous polyposis coli or *Apc;* Kirsten rat sarcoma virus or *Kras*) were synthesized by Sigma Laboratories (Aldrich, Mexico). The qPCR reaction was performed in 96-well PCR plates using 3.4 μL nuclease-free water, 5 μL SYBR^®^ Select Master Mix for CFX (Cat. No. 4472942, Applied Biosystems, Foster City, CA, USA), and 1 μL of cDNA template for each reaction. A BioRad thermocycler (CFX96 model C1000, Bio-Rad Laboratories, Inc., Hercules, CA, USA) was used under the following conditions: 95 °C for 10 min (15 s at 95 °C, 30 s at 60 °C, 30 s at 72 °C) for 35 cycles and then kept at 16 °C until the determination was made. For the analysis of the relative expression of the genes, it reported the threshold cycle of each gene, which is equivalent to the number of necessary cycles for each curve to reach a threshold in the fluorescence signal (2^−ΔΔCt^), using GAPDH as a housekeeping gene.

### 4.5. In Silico Assessment of the Interaction between Phenolic Compounds and Target Proteins

An in silico molecular docking analysis was performed to assess the potential of identified phenolic compounds quantified by HPLC-DAD from the extracts and the proteins codified for each quantified gene. The 3D structures of each phenolic compound were downloaded from the PubChem Database for: Apigenin (PubChem CID: 5280442), caffeic acid (PubChem CID: 689043), (+)-catechin (PubChem CID: 9064), chlorogenic acid (PubChem CID: 1794427), *p-*coumaric acid (PubChem CID: 637542), epigallocatechin gallate (PubChem CID: 65064), epicatechin (PubChem CID: 72276), ferulic acid (PubChem CID: 445858), gallic acid (PubChem CID: 370), 4-hydroxybenzoic acid (PubChem CID: 135), 4-hydroxyphenylacetic acid (PubChem CID: 127), luteolin (PubChem CID: 5280445), quercetin (PubChem CID: 5280343), resveratrol (PubChem CID: 445154), rosmarinic acid (PubChem CID: 5281792), rutin (PubChem CID: 5280805), sinapic acid (PubChem CID: 637775), and vanillin (PubChem CID: 1183). Each structure was converted to Protein Databank (PDB) files using Discovery Studio Visualizer v. 19.1.0.18287 (Dassault Systèmes, Vélizy-Villacoublay, France). As protein receptors, the 3D structures of APC (3NMZ) and K-RAS (6MQZ) proteins were also downloaded from the Protein Databank. Water molecules were removed from the proteins, and structures were prepared and docked using AutoDock Tools v. 1.5.6 [54]. Visualization of the best poses was carried out in Discovery Studio Visualizer.

### 4.6. Statistical Analysis

Data were presented as the means ± SD of at least two independent experiments in triplicate. Analysis of Variance (ANOVA) was conducted for the assays. For the phenolic compounds’ quantification, a post hoc Student’s *t*-test was conducted, whereas the in vitro assay tests were evaluated post hoc by the Tukey–Kramer test, using *p* < 0.05 as the significance parameter. The Principal Components Analysis (PCA) was conducted using the JMP v. 16 software.

## 5. Conclusions

The results suggest CA and PR extracts are polyphenol-rich edible plants with antiproliferative effects in vitro. The extracts displayed a pro-apoptotic impact, cell cycle arrest, increased tumor suppressor gene expression, and decreased selected oncogenes expression in the signaling pathways associated with colon carcinogenesis. Therefore, the aqueous extracts of CA and PR exert appropriate effects through the composition of their bioactive constituents. These results indicate the need for future research to investigate these extracts individually or in synergistic combination in colorectal cancer in vitro with more cell lines. The results obtained in the present study and further investigation following in vivo studies may provide further insight into the additional mechanisms involved in CRC treatment. In addition, the phytochemical composition could serve for further research to develop CA- and PR-enriched food products that show health benefits.

## Figures and Tables

**Figure 1 plants-12-01987-f001:**
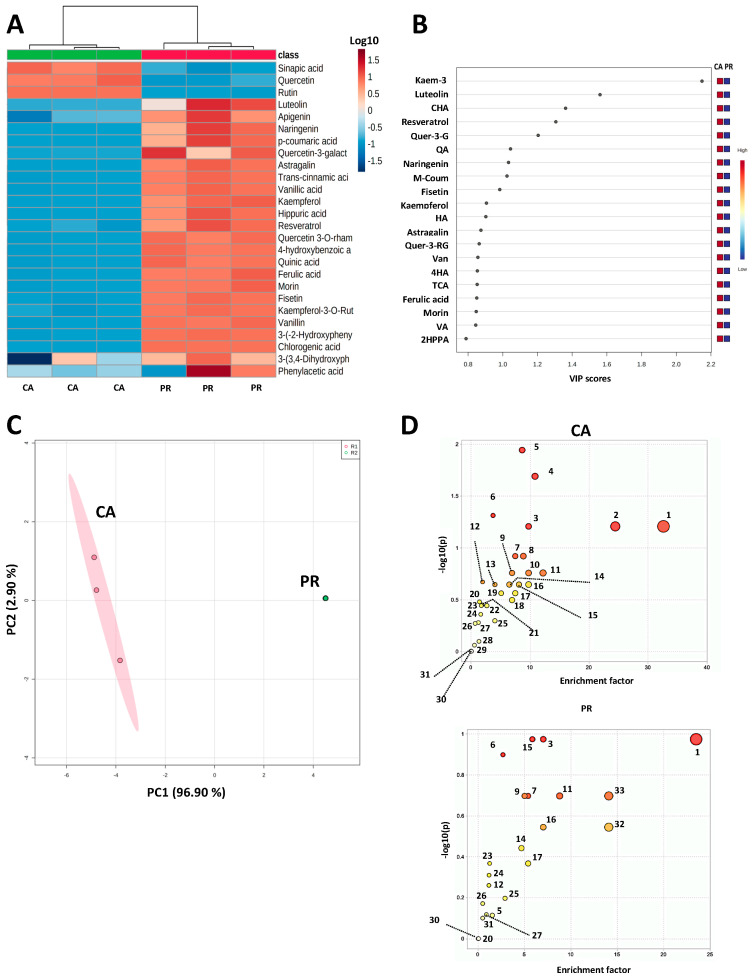
Untargeted metabolomic analysis of *Cnidoscolus aconitifolius* (CA) and *Porophyllum ruderale* (PR) phenolic compounds identified by UPLC-QTOF-MS-ESI. (**A**) Heatmaps depicting the normalized (Log10) relative abundance of each identified compound; (**B**) Variable in importance (VIP) scores of the identified phenolic compounds; (**C**) Principal component analysis (PCA) of the CA and PR extracts; (**D**) Mummichog pathways and network analysis of CA and PR extracts according to their metabolites’ profile. 2HPPA: 2-hydroxyphenylacetic acid; 4HA: 4-(3,4-dihydroxyphenyl)-propionic acid; CHA: Chlorogenic acid; HA: Hippuric acid; Kaem-3: Kaempferol-3-O-rutinoside (nicotiflorin); m-Coum: m-Coumaric acid; PC: Principal component; QA: Quinic acid; Quer-3-G: Quercetin-3-glucoside; Quer-3-RG: Quercetin-3-O-rhamnosyl-galactoside; TCA: Trans-cinnamic acid; VA: Vinyl acid; Van: Vanillin. 1: Indolizidines; 2: Quinoxalines; 3: Organic sulfuric acids; 4: Organic dicarboxylic acids; 5: Benzopyrans; 6: Pyridines; 7: Indolyl carboxylic acids; 8: Anilines; 9: Alkyl fluorides; 10: Benzoxazines; 11: Pteridines; 12: Benzenes; 13: Phosphate esters; 14: Anilides; 15: Organic thiophosphoric acids; 16: Benzodioxioles; 17: Benzoquinolines; 18: Lactones; 19: Imidazoles; 20: Organooxygen compounds; 21: Phenols; 22: Alcohols and polyols; 23: Organonitrogen compounds; 24: Carboxylic acids; 25: Anthracenes; 26: Pyrimidines; 27: Purines; 28: Benzoic acids; 29: Monosaccharides; 30: Benzamides; 31: Amino acids and peptides; 32: Dialkyl eters; 33: Phenylpiperidines.

**Figure 2 plants-12-01987-f002:**
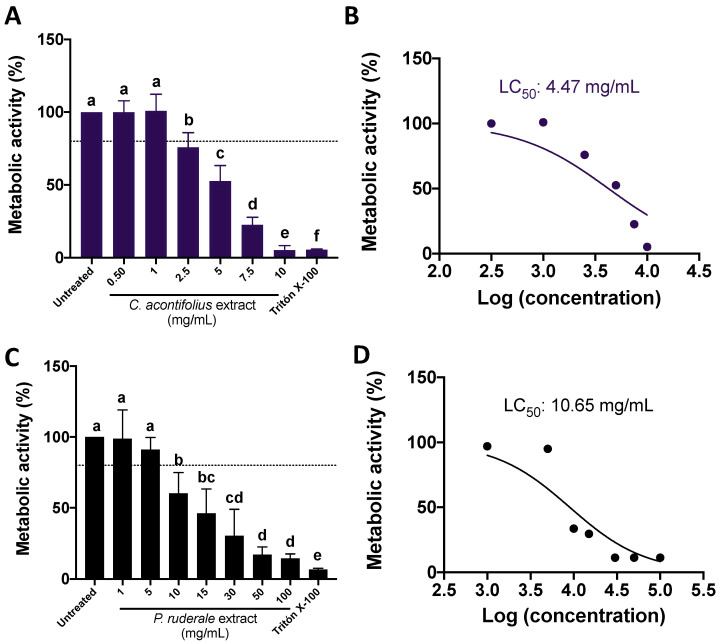
Impact of CA and PR extracts in the SW480 metabolic activity. (**A**) Effect of CA extracts (mg/mL) in the metabolic activity; (**B**) Adjusted dose–response equation and half-lethal dose (LC_50_) calculation from CA extracts; (**C**) Effect of PR extracts (mg/mL) in the cell’s metabolic activity; (**D**) Adjusted dose–response equation LC_50_ calculation from PR extracts. The results are expressed as the mean ± SD of three independent experiments in triplicate. Different letters represent significant differences (*p* < 0.05) by the Tukey–Kramer’s test. The dashed line from (**A**,**C**) indicates 80% metabolic activity. Untreated cells corresponded to untreated (DMEM + 0.5% bovine seric albumin, BSA) cells. Triton X-100 was used as a positive control. CA: *C. aconitifolius*; PR: *P. ruderale*.

**Figure 3 plants-12-01987-f003:**
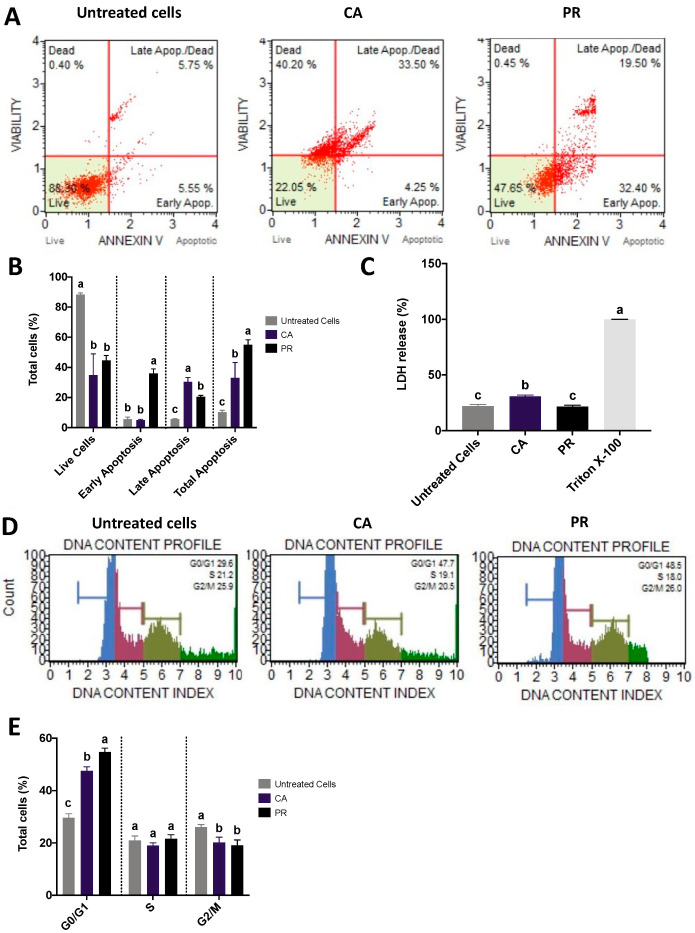
Assessment of cell death and lactate dehydrogenase release after treatment of SW480 cells with LC_50_ doses of CA and PR extracts. (**A**) Representative flow cytometry pictures of the impact of CA and PR LC_50_ doses on SW480 apoptosis; (**B**) Quantification of the total (%) live, early apoptotic, late apoptotic, and total apoptotic cells; (**C**) Lactate dehydrogenase (LDH) release; (**D**) Representative pictures of the impact of each treatment on cell cycle; (**E**) Quantification of the total cells (%) for each cell cycle. The results are expressed as the mean ± SD of three independent experiments in triplicate. Different letters express significant differences (*p* < 0.05) by the Tukey–Kramer’s test. For (**B**,**D**), statistical evaluation was conducted between treatments for each classification of cell death. For (**C**), statistical evaluation was completed among all groups. Untreated cells corresponded to untreated (DMEM + 0.5% BSA) cells. CA: *C. aconitifolius* extract (LC_50_: 5.28 mg/mL); PR: *P. ruderale* extract (LC_50_: 15.34 mg/mL). For the LDH assay, Triton X-100 was used as a positive control.

**Figure 4 plants-12-01987-f004:**
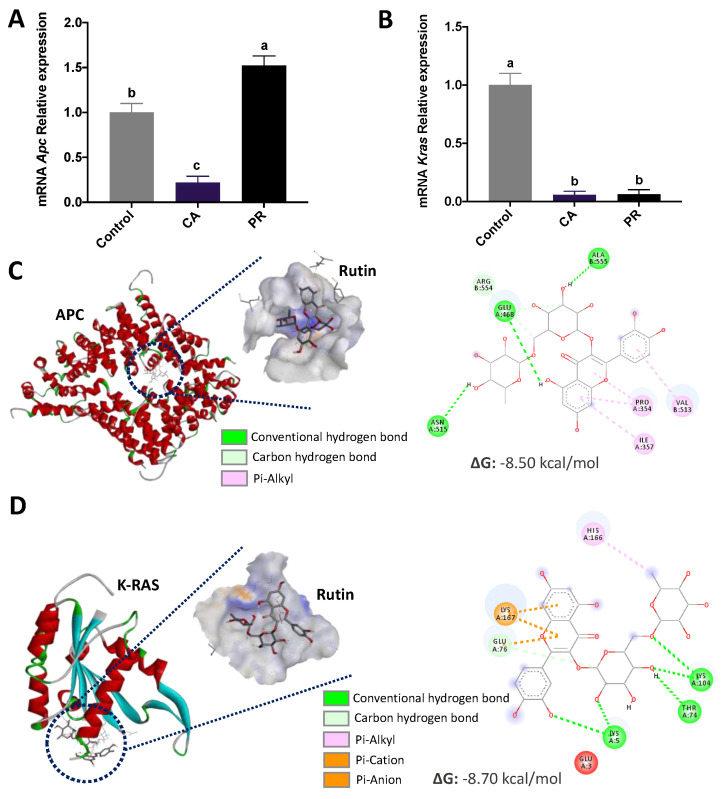
Evaluation of the effect of CA and PR treatments on *Apc* and *Kras* gene expression. (**A**) Average mRNA expression for the *Apc* gene of the SW480 cells treated with CA and PR extracts; (**B**) Average mRNA expression for the *Kras* gene of the SW480 cells treated with CA and PR extracts; (**C**) In silico assessment of the potential binding interaction between Rutin and the APC protein; (**D**) In silico evaluation of the potential binding interaction between Rutin and the K-RAS protein. The results in (**A**,**B**) are expressed as the mean ± SD of three independent experiments in triplicate. Different letters express significant differences (*p* < 0.05) by the Tukey–Kramer’s test. ΔG: Gibbs’ energy; CA: *C. aconitifolius*; PR: *P. ruderale*.

**Figure 5 plants-12-01987-f005:**
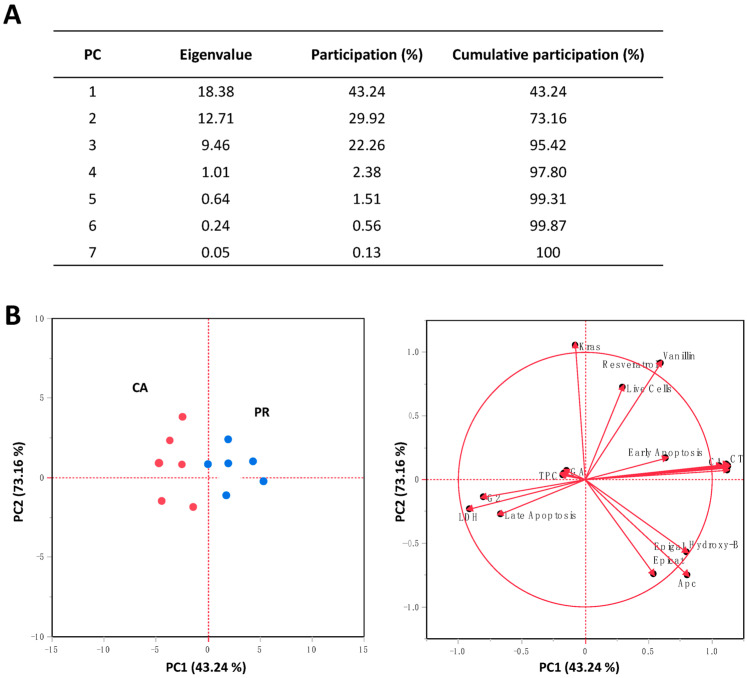
Principal Component Analysis (PCA) of CA and PR extracts, clustered by their effects on the evaluated cell parameters and the quantified HPLC-DAD compounds. (**A**) Eigenvalues and their participate percentage within the total variation of the analyzed variables; (**B**) Scatter plot and variables factor map. CA: *C. aconitifolius*; CT: Condensed tannins; LDH: Lactate dehydrogenase; Epicat: Epicatechin; Epigal: Epigallocatechin gallate; GA: Gallic acid; Hydroxy-B: Hydroxybenzoic acid; PC: Principal component; PR: *P. ruderale*; TPC: Total phenolic compounds.

**Table 1 plants-12-01987-t001:** Spectrophotometric determination of total phenolics (TPC), total flavonoids (TFC), and condensed tannins (CT) from CA and PR extracts.

Phenolic Compounds	CA	PR
TPC (mg GAE/100 g LE)	5109.00 ± 1034.00 *	2645.50 ± 10.70 *
TFC (mg CE/100 g LE)	309.50 ± 29.00 *	475.40 ± 26.10 *
CT (mg CE/100 g LE)	2.40 ± 0.10 *	3.00 ± 0.10 *

Results are expressed as the mean ± SD of three independent measurements in triplicate. Asterisks indicate significant differences (*p* < 0.05) by Student’s *t*-test. CA: *C. aconitifolius*; CE: (+)-catechin equivalents; CT: Condensed tannins; GAE: Gallic acid equivalents; LE: Lyophilized extract; PR: *p. ruderale*; TFC: Total flavonoids content; TPC: Total phenolic compounds.

**Table 2 plants-12-01987-t002:** Quantification of phenolic compounds of CA and PR by HPLC-DAD.

Compound Name	RT (min)	CA (mg eq./g DW)	PR (mg. eq./g. DW)
*Hydroxycinnamic acids and derivatives*			
Chlorogenic acid	10.58	6.51 ± 0.01 *	0.22 ± 0.01 *
Sinapic acid	11.67	3.87 ± 0.02 *	2.82 ± 0.06 *
Caffeic acid	11.82	14.46 ± 0.27 *	1.45 ± 0.01 *
*p-C*oumaric acid	14.42	6.36 ± 0.03 *	1.26 ± 0.13 *
Ferulic acid	15.30	5.17 ± 0.05 *	2.78 ± 0.02 *
*Hydroxybenzoic acids and derivatives and benzaldehydes*			
Gallic acid	7.49	2.04 ± 0.01 *	0.71 ± 0.02 *
Hydroxybenzoic acid	12.35	0.60 ± 0.01	0.11 ± 0.01
*Benzenoids*			
Hydroxyphenylacetic acid	9.89	1.05 ± 0.07 *	0.11 ± 0.02 *
*Flavonols*			
Rutin	12.90	1.22 ± 0.18 *	2.83 ± 0.14 *
Quercetin	18.44	0.20 ± 0.01	0.14 ± 0.01
*Flavones*			
(+)-catechin	10.85	0.45 ± 0.01 *	14.01 ± 0.04 *
Epicatechin	11.55	1.06 ± 0.1 *	57.74 ± 0.29 *
Epigallocatechin gallate	11.80	3.29 ± 0.01 *	14.37 ± 0.04 *
*Other compounds*			
Resveratrol	17.46	3.62 ± 0.31 *	0.02 ± 0.01 *

Results are expressed as the mean ± SD of two independent experiments in triplicate. Asterisks indicate significant differences by Student’s *t*-test (*p* < 0.05). CA: *C. aconitifolius* extract; DW: Dry weight; PR: *P. ruderale* extract. Text in italics indicate the phenolic compounds groups.

## Data Availability

Data will be available upon reasonable request.

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
