# Peer review of "Comparison of Phytochemical Composition and Untargeted Metabolomic Analysis of an Extract from Cnidoscolus aconitifolius (Mill.) I. I. Johnst and Porophyllum ruderale (Jacq.) Cass. and Biological Cytotoxic and Antiproliferative Activity In Vitro"

_plants, 2023, doi:10.3390/plants12101987_

Round 1
Reviewer 1 Report
The presented manuscript covers a thorough investigation on the composition and cytotoxicity of extract from Cnidoscolus aconitifolius. In this manuscript, the authors identified and quantified the major compounds in CA and PR extracts followed by evaluation of CA/PR extracts in SW480 cells for cytotoxicity and antiproliferative effect. Overall, this is an interesting research article, revealing the potential of CA extracts in cancer treatments, which would attract wide interest in this field. Here attached my comments.
1. The authors did a great job in identification of the major compounds in CA extracts, it would be more helpful if the authors could include a representative Mass Spec results in the manuscript or in supplemental files.
2. Although the authors include a lot of information in the discussion section, it would be better to reorganize the content. I would suggest stating the rationale of this study, why choose colon cancer and the importance of Apc and Kras involved in CRC in the background section to clarify the aim of the study. Also, the explanation of cell cycle should also be included in the corresponding results section to illustrate what it means by arresting cells in certain stages.
3. In the discussion section, the authors mentioned that CA extracts show cytotoxicity effect in other cancer cell lines. Corresponding citations need to be listed.
4. Is there any study of CA extracts in vivo? Could the authors provide a brief discussion about the in vivo fate of CA extracts?
There are several typos need to be fixed. Otherwise, the English language is good.
Author Response
Response to Reviewer 1 Comments
Reviewer: The presented manuscript covers a thorough investigation on the composition and cytotoxicity of extract from Cnidoscolus aconitifolius. In this manuscript, the authors identified and quantified the major compounds in CA and PR extracts followed by evaluation of CA/PR extracts in SW480 cells for cytotoxicity and antiproliferative effect. Overall, this is an interesting research article, revealing the potential of CA extracts in cancer treatments, which would attract wide interest in this field. Here attached my comments.
Authors’ response: We appreciate the reviewer’s comments.
Point 1: The authors did a great job in identification of the major compounds in CA extracts, it would be more helpful if the authors could include a representative Mass Spec results in the manuscript or in supplemental files.
Response 1: Thanks. Representative chromatograms of the identified compounds in both electrospray ionization modes were added as Supplementary Figure S1 and Supplementary Figure S2. Please refer to the revised Manuscript and the revised Supplementary Material.
- Revised manuscript:
Page 3, Lines 100-103: “(…) some of them confirmed in Supplementary Table S1. Representative chromatograms of some of the identified compounds in the positive and negative electrospray ionization (ESI) are presented in Supplementary Figure S1 and Supplementary Figure S2, respectively.”
Point 2: Although the authors include a lot of information in the discussion section, it would be better to reorganize the content. I would suggest stating the rationale of this study, why choose colon cancer and the importance of Apc and Kras involved in CRC in the background section to clarify the aim of the study. Also, the explanation of cell cycle should also be included in the corresponding results section to illustrate what it means by arresting cells in certain stages.
Response 2: Thanks. The rationale of the study and why choosing colon cancer, and the importance of Apc and Kras in CRC was included as suggested.
- Revised manuscript:
Pages 8-9, Lines 238-257: “Cancer is a multifactorial heterogeneous metabolic pathology, defined as an irreversible alteration in cell homeostasis [10]. Colorectal cancer (CRC) is currently the third leading cause of cancer death, and global mortality cases are estimated to increase by 60% by the year 2030 [11]. Although several treatments have been proposed for CRC, most of these procedures involving pharmacological methods display several side effects that could be reduced with the involvement of natural products aiming to either reduce the effect of these treatments or diminish the risk of having CRC [12]. Genetic and epigenetic alterations are involved in CRC as well as mutations that inactivate the function of the Apc gene and the Ras oncogene, mainly in KRAS. The latter oncogene contains membrane-associated-GTPases that are involved in cell survival, proliferation, and differentiation. In CRC, mutations in the KRAS oncogene and those of the APC gene lead to greater cell proliferation, the MAPK signaling caused by mutated KRAS presents ERK hyperphosphorylation, which activates different effector mechanisms such as the G1/S phase transition, and the inhibition of apoptosis. The defective APC coupled with the mutated KRAS inhibits the activity of GSK3β (glycogen synthase kinase) and amplifies the activity of β-catenin [13].
Among the natural treatments that could be used as a chemopreventive approach in CRC treatment, natural plants have emerged as a hot topic due to the revalorization of their phytochemical composition, exhibiting various biochemical mechanisms inhibiting CRC progression [14]”
Page 11, Lines 360-363: “The normal cell cycle develops according to the G1-S-G2-M phases, and in this cellular mechanism, the synthesis and distribution of nucleic acids, chromosomes and proteins are verified. At an abnormal point, the cell cycle proceeds disorderly, causing alterations in cell proliferation and differentiation [33].”
Page 11, Lines 368-370: “Before the G0/G1 arrest, rapid induction of cyclin-dependent kinase inhibitors occurs, leading to phosphorylation of pocket protein family members, inhibiting cell cycle progression [35].”
Point 3: In the discussion section, the authors mentioned that CA extracts show cytotoxicity effect in other cancer cell lines. Corresponding citations need to be listed.
Response 3: Thanks. Cytotoxicity was evaluated in several breast and lung cancer cell lines. The reference was added in the revised manuscript.
- Revised manuscript:
Page 10, Lines 327-328: “although biological features of the tested cell lines could explain these differences [28].”
Ikpefan, E.O.; Ayinde, B.A.; Mudassir, A.; Farooq, A.D. Comparative in vitro Assessment of the Methanol Extracts of the Leaf, Stem, and Root Barks of Cnidoscolus Aconitifolius on Lung and Breast Cancer Cell Lines. Turkish J. Pharm. Sci. 2019, 16, 375–379, doi:10.4274/tjps.galenos.2018.19942.
Point 4: Is there any study of CA extracts in vivo? Could the authors provide a brief discussion about the in vivo fate of CA extracts?
Response 4: Thanks. So far, there is only 1 study reporting the potential chemopreventive effect of CA extract in early colorectal cancer in vivo. Particularly for CA.
We have added and expanded the discussion of this report. Please refer to the Revised manuscript.
- Revised manuscript:
Page 10, Lines 308-322: “Still, our research group previously informed the impact of C. aconitifoliusconsumption in an azoxymethane-induced early colorectal cancer in vivo model [9]. In this research, colon cancer was induced using azoxymethane, a common carcinogen, in male Sprague-Dawley rats, and CA was administered to the rats as an aqueous extract (10 g of the leaves boiled in 1 L water for 5 min). Despite no differences were shown for the body weight evolution of the animals between the tested groups, a significant reduction (-29.5 to -64.6 %) in colonic aberrant crypt foci in a chronic and sub-chronic administration of the leaves (16 and 32 weeks, respectively) was found, suggesting the effect of C. aconitifolius bioactive components in reducing the colonic histological alteration. No individual compounds from CA were evaluated in this model, but the CA phytochemical composition, predominantly governed by p-coumaric, rosmarinic, and chlorogenic acids, was linked to the chemopreventive effects in reducing the immunohistochemical expression of proteins in the colonic tissue linked to CRC progression and development such as b-catenin, proliferating cell nuclear antigen (PCNA), caspase-3, cyclo-oxygenase-2 (COX-2), and the nuclear factor kappa B (NF-kB).
Kuri-García, A.; Godínez-Santillán, R.I.; Mejía, C.; Ferriz-Martínez, R.A.; García-Solís, P.; Enríquez-Vázquez, A.; García-Gasca, T.; Guzmán-Maldonado, S.H.; Chávez-Servín, J.L. Preventive Effect of an Infusion of the Aqueous Extract of Chaya Leaves (Cnidoscolus Aconitifolius) in an Aberrant Crypt Foci Rat Model Induced by Azoxymethane and Dextran Sulfate Sodium. J. Med. Food 2019, 22, 851–860, doi:10.1089/jmf.2019.0031.
Point 5: There are several typos need to be fixed. Otherwise, the English language is good.
Response 5: Thanks. We have double-checked the manuscript and the typos were fixed. Please refer to the Revised Manuscript.
- Revised manuscript:
Page 1, Lines 32-33: “Results suggested that CA and PR are polyphenol-rich plant sources exhibiting antiproliferative effects in vitro.”
Page 2, Line 51: “(…) such as cancer, arthritis, diabetes mellitus, and gastrointestinal (…)”.
Page 2, Line 52: “Moreover, PR is used as an anti-inflammatory treatment (…)”.
Page 2, Lines 54-55: “(…) the treatment of gastric conditions”
Page 2, Line 58-59: “(…) The CA food and medicinal value have been (…)”
Page 2, Line 62: “In contrast, leaf extracts have been (…)”.
Page 2, Lines 74-75: “Table 1 shows the total amount of free total phenolic compounds (TPC), total flavonoid content”
Page 2, Line 85: “Supplementary Table S1 shows”
Page 4, Line 124: “dialkyl ethers, amino acids”.
Page 9, Line 270: “but higher than the values”.
Page 9, Line 272: “Regarding PR, TPC contents reported”
Page 9, Line 285: “among P. ruderale main phenolics, it could be found”
Page 10, Lines 329-330: “but even less information on the anti-cancer potential of P. ruderale is found in the literature”.
Page 11, Line 379: “APC Wnt/b-catenin and K-RAS”.
Page 11, Line 392: “which might be inferred”.
Page 12, Line 406: “accessed on the 2nd of July/2022”.
Page 12, Line 413: “screened through a 0.5 mm particle-size sieve”
Page 12, Line 449: “for the positive scan mode) was set.”.
Page 13, Line 575: “resveratrol, and rutin.”
Page 13, Line 502: “After incubation, the medium”.
Page 14, Line 544: “Each 2 μg of mRNA was converted”.
Page 15, Line 561: “equivalent to the number of necessary”.
Page 15, Line 586: “Analysis of Variance (ANOVA) was conducted”.
Thank you very much for your review.
"Please see the attachment."

Reviewer 2 Report
In this work, Angel Felix et al disclosed the phytochemical compositions and untargeted metabolomics analysis of C. aconitifolius and P. ruderale, as well as their anti-proliferative activities. The manuscript is well prepared. And some interesting results were provided based on their investigations. However, this reviewer is not convinced at present stage since the work is too preliminary. I would consider the acceptance for the publication in Plants after the substantial improvements.
Major points:
1) The work seems to be a simple combination of several pieces of loosely-connected studies. The logic connections between chemical and biological studies, such as chemical constituents—KRAS/APC—cell cycle—apoptosis—cell viability, are quite forced. There is no solid evidence to support the conclusions that the authors came to in almost every section. The authors just touched several significantly varied aspects of CA and PR. The reviewer suggested that the author focus on some limited aspects of the medicines and provides solid and convincing evidence to come to a careful conclusion.
2) Some results that are not very critical and informative may be moved from manuscript to the Supplementary Information, such as table 2 in page 3.
3) It’s suggested to remove the contents concerning the necrosis in 2.3 part page 7, since there is no results related to this topic.
Author Response
Response to Reviewer 2 Comments
Reviewer: In this work, Angel Felix et al disclosed the phytochemical compositions and untargeted metabolomics analysis of C. aconitifolius and P. ruderale, as well as their anti-proliferative activities. The manuscript is well prepared. And some interesting results were provided based on their investigations. However, this reviewer is not convinced at present stage since the work is too preliminary. I would consider the acceptance for the publication in Plants after the substantial improvements.
Authors' response: We appreciate the reviewer’s comments, and we hope the amendments done in the manuscript could make it suitable for publication in Plants.
Point 1: The work seems to be a simple combination of several pieces of loosely connected studies. The logic connections between chemical and biological studies, such as chemical constituents—KRAS/APC—cell cycle—apoptosis—cell viability, are quite forced. There is no solid evidence to support the conclusions that the authors came to in almost every section. The authors just touched several significantly varied aspects of CA and PR. The reviewer suggested that the author focus on some limited aspects of the medicines and provides solid and convincing evidence to come to a careful conclusion.
Response 1.1: We appreciate the reviewer’s comments. Our research group previously conducted an in vivostudy evaluating the impact of CA extract consumption, and we did not find differences in the protein expression of several proteins linked to CRC progression and development such as b-catenin, PCNA, caspase-3, COX-2, and NF-kB. However, since we observed a reduction in the colonic aberrant crypt foci number, and a protective histological effect of CA in those tissues from animals treated with a carcinogen (azoxymethane), we decided to turn back to an in vitro study to assess additional mechanisms that could be associated to CRC prevention. Moreover, CA and Porophyllum ruderale are some of the most consumed “Quelites” in Mexico, and there are no studies validating their biological effects, at least for CRC, and based on this we decided to also include P. ruderale extracts in the evaluation to also assess their biological effect. We have added this information in the Revised Manuscript.
- Revised manuscript:
Page 10, Lines 308-324: “Still, our research group previously informed the impact of C. aconitifolius consumption in an azoxymethane-induced early colorectal cancer in vivo model [9]. In this research, colon cancer was induced using azoxymethane, a common carcinogen, in male Sprague-Dawley rats, and CA was administered to the rats as an aqueous extract (10 g of the leaves boiled in 1 L water for 5 min). Despite no differences were shown for the body weight evolution of the animals between the tested groups, a significant reduction (-29.5 to -64.6 %) in colonic aberrant crypt foci in a chronic and sub-chronic administration of the leaves (16 and 32 weeks, respectively) was found, suggesting the effect of C. aconitifolius bioactive components in reducing the colonic histological alteration. No individual compounds from CA were evaluated in this model, but the CA phytochemical composition, predominantly governed by p-coumaric, rosmarinic, and chlorogenic acids, was linked to the chemopreventive effects in reducing the immunohistochemical expression of proteins in the colonic tissue linked to CRC progression and development such as b-catenin, proliferating cell nuclear antigen (PCNA), caspase-3, cyclo-oxygenase-2 (COX-2), and the nuclear factor kappa B (NF-kB). Yet, no differences were found between the CA and AOM+CA treatments, suggesting that additional mechanisms could be explored to explain the chemoprotective effects of CA.”
- Response 1.2: The genes we have presented in this research were selected based on the consideration that these genes are critical in CRC development. We have included this in the Discussion section. Please refer to the revised manuscript.
- Revised manuscript:
Pages 8-9, Lines 244-253: “Genetic and epigenetic alterations are involved in CRC as well as mutations that inactivate the function of the Apc gene and the Ras oncogene, mainly in KRAS. The latter oncogene contains membrane-associated-GTPases that are involved in cell survival, proliferation, and differentiation. In CRC, mutations in the Keas oncogene and those of the Apc gene lead to greater cell proliferation, the MAPK signaling caused by mutated KRAS presents ERK hyperphosphorylation, which activates different effector mechanisms such as the G1/S phase transition, and the inhibition of apoptosis. The defective APC coupled with the mutated KRAS inhibits the activity of GSK3β (glycogen synthase kinase) and amplifies the activity of β-catenin [13].”
László, L.; Kurilla, A.; Takács, T.; Kudlik, G.; Koprivanacz, K.; Buday, L.; Vas, V. Recent Updates on the Significance of KRAS Mutations in Colorectal Cancer Biology. Cells 2021, 10, 667, doi:10.3390/cells10030667.
- Response 1.3: We agree with the reviewer that additional studies are required to gather more definitive conclusions from the results. We have added information about this in the revised manuscript.
Revised manuscript:
Page 10, Lines 301-302: “more research is needed to elucidate their phytochemical composition aiming for future biotechnological applications.”
Page 11, Lines 355-359: “It is worth noting that apoptosis is a complex process involving the activation of several proteins and biochemical mechanisms that cannot be wholly elucidated through the cell cycle and apoptosis measurement through colorimetric or fluorometric methods. However, the provided quantifications from this research open the possibility of further exploring these mechanisms through additional proteomics and gene expression analysis”
Page 11, Lines 396-398: “The individual evaluation of these components directly extracted from CA and PR might conduct to explain the contributions of phytochemicals to CRC development in vitro and in vivo.”
Page 15, Lines 595-597: “Further research exploring these extracts in CRC in vitro with more cell lines and in vivowill provide additional mechanisms involved in CRC reduction and treatment.”
Point 2: Some results that are not very critical and informative may be moved from manuscript to the Supplementary Information, such as table 2 in page 3.
Response 2: Thanks. We have removed the information from Table 2 from the main manuscript and moved it as Supplementary Information. Please refer to the Revised Manuscript and Revised Supplementary Information.
- Revised manuscript:
Page 3, Lines 100-103: “some of them confirmed in Supplementary Table S1. Representative chromatograms of some of the identified compounds in the positive and negative electrospray ionization (ESI) are presented in Supplementary Figure S1 and Supplementary Figure S2, respectively.”
Point 3: It’s suggested to remove the contents concerning the necrosis in 2.3 part page 7, since there is no results related to this topic.
Response 3: Thanks. Information indicating “necrosis” was removed and instead, LDH release was presented. Please refer to the Revised manuscript.
- Revised manuscript:
Page 5, Line 161: “Examination of the SW480 cell death and lactate dehydrogenase release after (…)”.
Page 5, Line 168: “For the lactate dehydrogenase release (Figure 3C)”.
Page 7, Line 182: “(C) Lactate dehydrogenase release”.
Page 10, Lines 337-343: “The ability of PA and CR extracts to induce apoptosis, induce LDH release, and arrest the cell cycle was confirmed in this research as the LC50 concentrations of the extracts significantly increased the number of cells in total apoptosis, but PR treatment was more effective. In contrast, CA was more cytotoxic, inducing higher levels of LDH release. Despite no reports found for CA and PR inducing LDH release and apoptosis, medicinal leaves have displayed the reduction of LDH release, an indicator of cell membrane damage [30]”.
Page 14, Line 509: “4.4.2. Determination of lactate dehydrogenase (LDH) release”.
Thank you very much for your review.
"Please see the attachment."

Round 2
Reviewer 1 Report
I would suggest acceptance in current version.
Author Response
Reviewer 1
- Reviewer: I would suggest the acceptance in current version
- Authors’ response: We appreciate the reviewer’s comments.
Reviewer 2 Report
In the updated manuscript, the authors just modified the presentation of the work, as well as some discussions. No significant improvements were made. This reviewer is still conserved on the present form of the work.
Author Response
We appreciate your valuable comments.
Please see the attachment.
Best regards
